# Active Manipulation of the Spin and Orbital Angular Momentums in a Terahertz Graphene-Based Hybrid Plasmonic Waveguide

**DOI:** 10.3390/nano10122436

**Published:** 2020-12-05

**Authors:** Ziang Wang, Qilong Tan, Yong Liang, Xia Zhou, Wen Zhou, Xuguang Huang

**Affiliations:** 1Guangzhou Key Laboratory for Special Fiber Photonic Devices and Applications, South China Normal University, Guangzhou 510006, China; 2018022060@m.scnu.edu.cn (Z.W.); 2018022045@m.scnu.edu.cn (Y.L.); XZhou@m.scnu.edu.cn (X.Z.); 2School of Physics and Telecommunications Engineering, South China Normal University, Guangzhou 510006, China; tanqilong@m.scnu.edu.cn; 3Department of Materials, University of Oxford, Oxford OX1 3PH, UK

**Keywords:** spin angular momentum, orbital angular momentum, graphene plasmonic waveguides

## Abstract

Angular momentums (AMs) of photons are crucial physical properties exploited in many fields such as optical communication, optical imaging, and quantum information processing. However, the active manipulation (generation, switching, and conversion) of AMs of light on a photonic chip remains a challenge. Here, we propose and numerically demonstrate a reconfigurable graphene-based hybrid plasmonic waveguide (GHPW) with multiple functions for on-chip AMs manipulation. Its physical mechanism lies in creating a switchable phase delay of ±π/2 between the two orthogonal and decomposed linear-polarized waveguide modes and the spin-orbit coupling in the GHPW. For the linear-polarized input light with a fixed polarization angle of 45°, we can simultaneously switch the chirality (with −*ħ*/+*ħ*) of the transverse component and the spirality (topological charge *ℓ* = −1/+1) of the longitudinal component of the output terahertz (THz) light. With a switchable phase delay of ±π in the GHPW, we also developed the function of simultaneous conversion of the charity and spirality for the circular-polarized input light. In addition, a selective linear polarization filtering with a high extinction ratio can be realized. With the above multiple functions, our proposed GHPWs are a promising platform in AMs generation, switching, conversion, and polarization filtering, which will greatly expand its applications in the THz photonic integrated circuits.

## 1. Introduction

The angular momentums of electromagnetic wave can be divided into spin angular momentum (SAM) and orbital angular momentum (OAM) [1]. SAM is determined by the circular polarization states. The eigenstate of a spinning single photon can be expressed as ± *ħ*, which represents the left and right handedness, respectively. Conventionally, the SAM can be produced by a quarter-wave plate, which converts linearly polarized light into circularly polarized light. To shrink down the footprint, metasurfaces were proposed as ultra-thin QWPs [2,3]. On the other hand, OAM is related to the helical phase of the wavefront, and each vortex photon can carry a discrete eigenstate of *ℓħ*, where *ℓ* is an integer. OAM can be generated with different approaches, including using a spiral phase plate [4], digital micromirror device [5], cylindrical lens mode converter [6], chirally-coupled core fiber (CCCF) [7], and more recently metasurfaces [8,9,10,11]. It is worth noting that the dynamic controlling of SAM and OAM is still a challenge on an integrated platform.

An integrated waveguide platform has the figure of merits of ultra-compact footprint, low loss, high integration density, and phase stability. Wave plates and chirality splitters were proposed based on the dielectric waveguide platforms [12,13,14]. Vertex beam emitters were proposed using the silicon photonic micro-ring resonator patterned with angular gratings [15] and a pixeled micron-sized disk with broad working bandwidth [16]. Noteworthy, previous works [17,18] show that SAM and OAM can be simultaneously generated and transmitted in the hybrid plasmonic waveguides in the telecommunication band using waveguide birefringence and spin-orbit coupling principles. However, they were only focused on the passive devices with fixed angular momentums of output photon depending on the designed structural parameters [19].

In this article, we propose a reconfigurable graphene-based hybrid plasmonic waveguide (GHPW) with four different functions including active manipulation (i.e., generation, switching, and converting) both SAM and OAM of photons and selective linear-polarization filtering in the THz spectral band for the first time. The working principle for the first two functions is based on the excitation of polarization-dependent hybrid plasmonic mode by tuning the Fermi level of graphene capacitors in the GHPW, which results in a strong waveguide birefringence with a switchable relative phase difference of ±π/2 between the quasi-TE and the quasi-TM modes. Therefore, a circularly polarized light with a switchable (left or right) handedness can be generated from a quasi-linearly polarized input light with a fixed polarization angle of 45°, while encoding OAM with switchable spirality into its longitudinal electric field based on the spin-orbit coupling of the photons in the waveguide. When the phase delay is ±π, the third function of simultaneous conversion of the charity and spirality is developed for the circular-polarized input light. In the same GHPW, selective linear-polarization filtering with a high extinction ratio can be realized as its fourth function. With versatile functionality, our work presents a novel modulation strategy for manipulating angular momentums in the proposed GHPW, offering new and exciting opportunities for fundamental research and practical applications that require dynamic controlling of the chirality (handedness) and spirality (topological charges) of photons in biomolecules sensing and on-chip quantum information processing [20,21,22].

## 2. Theoretical Model

As shown in Figure 1a,b, when a 45° linearly polarized light (*α* = 45°) is coupled into the graphene hybrid plasmonic waveguide device, the spin angular momentum (SAM) and orbital angular momentum (OAM) can be encoded onto the transverse electric field component (*E*_T_) and longitudinal component (*E*_Z_) of the output beam, respectively. The proposed waveguide is mainly composed of three waveguide sections. The first and the third waveguide sections have the same structure, using gallium arsenide (GaAs) as the waveguide core with a refractive index of 3.6 and with a height and width of 22 μm, which is surrounded by a layer of high-density polyethylene (SiO_2_) with a low refractive index of 1.54. Note that GaAs and SiO_2_ are widely used in the terahertz (THz) spectral range [23,24]. When the THz light wave is coupled into the waveguide from a lens fiber, it requires a certain transition distance to form a stable eigenmode, and the length of the first waveguide section is set as 50 μm. Similarly, when the THz light wave is emitted from the second waveguide section (GHPW), a 50-μm-long transition is required to form a stable eigenmode in the pure dielectric waveguide (third waveguide section). In addition, the operating wavelength in the work is λ = 99.93 μm (3THz).

In these two pure dielectric waveguides, the calculated effective refractive indices of the quasi-TE mode and the quasi-TM mode are 2.45, so there is no phase difference due to symmetric cross sections of the first and third waveguides. The second waveguide section is a graphene-based hybrid plasmonic waveguide, in which SAM and OAM are respectively encoded in the transverse and longitudinal components (*E*_Z_) of the electric field, and which can be actively manipulated. As shown in Figure 1, five layers of graphene are laid under and on the sidewalls of the GaAs waveguide respectively, and the thickness of the SiO_2_ layer is 4 μm. Two polysilicon films with relative permittivity *ε*_p_ = 3 and thickness *g* = 0.5 μm were deposited and sandwiched by the SiO_2_ layers. The center-to-center separation between the polysilicon and graphene layer is 1 μm. Gate voltage *V*_g_ is applied to the graphene-SiO_2_ spacer-polysilicon parallel plate capacitor to control the conductivity of graphene via the electrostatic doping effect, as illustrated in Figure 1a,b. Especially, graphene supports surface plasmons in terahertz and infrared wavelengths [25], which has a stronger mode confinement and relatively smaller transmission loss than noble metals, with remarkable advantages of active tunability for photonic integrated circuits [26,27]. Note that the polysilicon layer is very thin with a moderate relative permittivity, it only introduces little perturbation on the excited mode profiles compared with those in the same waveguide structure without polysilicon. It is worth mentioning that a possible fabrication scheme for the hybrid waveguide could be seen in the Appendix A.

In this study, we use the commercial software Lumerical FDTD Solutions to numerically calculate and analyze the properties of the proposed GHPW. The 2D surface model is employed to describe the monolayer graphene [28,29]. The graphene’s electromagnetic properties are characterized by a surface conductivity *σ_g_*, which can be calculated by the Kubo formula under the local random phase approximation (RPA) [30]:(1)$$\begin{array}{l} \sigma_{g}(\omega ,\tau ,T,E_{f})=\frac{2ie^{2}k_{B}T}{\pi \hbar^{2}(\omega +i\tau^{-1})}\ln[2\cosh(\frac{E_{f}}{2k_{B}T})]\\ +\frac{e^{2}}{4\hbar }\{\frac{1}{2}+\frac{1}{\pi }\arctan(\frac{\hbar \omega -2E_{f}}{2k_{B}T})-\frac{i}{2\pi }\ln[\frac{{\left(\hbar \omega +2E_{f}\right)}^{2}}{{\left(\hbar \omega -2E_{f}\right)}^{2}+{\left(2k_{B}T\right)}^{2}}]\} \end{array}$$

Here, *ħ*, *ω*, *k_B_*, *T* (= 300 K), *τ* (= 0.5 ps) are the reduced Planck’s constant, radian frequency, Boltzmann constant, room temperature, and carrier relaxation time, respectively. The Fermi level of graphene can be calculated by *E_f_* = *ħv_f_* (*πn_g_*)^1/2^. *v_f_* and *n_g_* are the Fermi velocity and carrier concentration, respectively. We set the Fermi level of graphene less than 0.9 eV. It also has been theoretically and experimentally demonstrated that *E_f_* can be easily tuned from −1.0 to 1.0 eV via the electrostatic gating. Therefore, the Fermi level range we set is practical. By taking an individual graphene sheet as a non-interacting monolayer, the optical conductivity of few-layer graphene is *Nσ_g_*, where *N* is the number of layers (*N* < 6). In addition, compared with monolayer graphene plasmons, few-layer graphene plasmons have the capability of effectively reducing the wavenumber of surface plasmon polariton (SPP), resulting in a much weaker confinement to the SPP mode [31]. In addition, for a small number of graphene layers with less than six layers, the absorption loss decreases with the increase of the number of layers. Therefore, we set the number of graphene layers as five in this study [32].

## 3. Results and Discussion

### 3.1. Waveguide Birefringence and the Spin-Orbit Coupling in the GHPWs

The proposed graphene-based hybrid plasmonic waveguide has a strong birefringent effect, i.e., the hybrid plasmonic mode strongly confined in the low-refractive-index SiO_2_ can be selectively excited for the quasi-TE or the quasi-TM mode. Therefore, we use this birefringent effect in the hybrid plasmonic waveguide to rotate the polarization of the photon and spiral its wavefront. Figure 2a shows a linear-polarized THz light wave with a polarization angle of 45° coupled into the first waveguide section (pure dielectric waveguide), which has a decomposed *x* component (the quasi-TE mode) and *y* component (the quasi-TM mode). In addition to the transverse electric field *E*_T_, there was a longitudinal electric field component *E*_Z_ generated along the light propagation direction due to the strong transverse limitation of the waveguide [33]. The well-known spin-orbit coupling shows that the light carrying OAM can be generated from light initially only carrying SAM. Mathematically, coupling between *E*_Z_ and *E*_T_ for this scenario can be expressed as [34]:(2)$$E_{z}=\frac{1}{i\beta }\nabla_{T}E_{T},$$
where ∇*_T_* and *β* are the transverse gradient and the propagation constant, respectively. The amplitude of longitudinal component (*E*_*Z*_) is decided by the geometric parameters of the waveguide [34]. In our work, the longitudinal component amplitudes (*E*_*Z*_) of TE and TM modes are both about 40% of their transverse component amplitudes (*E*_*T*_). The optical field of the fundamental mode at the center of GaAs square waveguide is purely transverse. Additionally, in our proposed waveguide, the higher-order modes of terahertz light wave with *λ* = 99.93 μm are cut-off. Here, we characterize the polarization state of the mode at the center of the waveguide. Furthermore, the polarization angle α of the input light can be defined as:(3)$$\tan(\alpha )=\left|\frac{E_{T-TM1}}{E_{T-TE1}}\right|$$ where |*E*_*T*-*TE*1_| and |*E*_*T*-*TM*1_| are respectively the amplitudes of the decomposed *x* and *y* components of *E**_T_* in the first waveguide section. Figure 2b shows that there is a significant difference in the modal field profiles for the quasi-TE and quasi-TM modes due to the selective excitation of the hybrid plasmonic mode when the THz light wave transmits to the second waveguide section (GHPW).

In the first case, the Fermi energy level of the bottom graphene is set as *E_f_* = 0.9 eV and the Fermi level of the sidewall graphene is 0 eV, a large difference between the effective refractive indices of the two orthogonal modes can be introduced, and is expressed as follows:(4)$$\Delta n=Re[n_{\mathrm{quasi}-\mathrm{TM}}-n_{\mathrm{quasi}-\mathrm{TE}}]$$

The calculated effective refractive index of the quasi-TE mode and the quasi-TM mode are *n*_quasi-TE_ = 2.386 + 0.0067i and *n*_quasi-TM_ = 2.565 + 0.0046i, respectively. The imaginary part of the effective index represents the propagation loss of the mode. Due to the small difference in the imaginary parts, here we only focus on the real part of the effective refractive index.

In the second waveguide section, with an effective refractive index difference Δ*n* = 0.179, it produces a relative phase difference *δ* between the quasi-TM mode and the quasi-TE mode, which can be expressed as:(5)$$\delta =k_{0}\Delta nL_{g}+\Delta \delta$$ where Δ*δ*, *L*_g_, and *k*_0_ are respectively the initial phase difference, length of graphene, and the wave vector in free space. When Δ*δ* = 0, for the THz light wave with wavelength *λ* = 99.93 μm, the theoretically calculated length of the bottom graphene is 135 μm to obtain Δ*δ* = π/2.

Similarly in the second case, when the Fermi level of the sidewall graphene is set as *E_f_* = 0.9 eV and the Fermi level of the bottom graphene is 0 eV, the effective refractive indices of the quasi-TE mode and the quasi-TM mode are respectively *n*_quasi-TE_ = 2.579 + 0.0059i and *n*_quasi-TM_ = 2.388 + 0.0061i, and the effective refractive index difference of the two modes is Δ*n* = 0.191, then the length of the sidewall graphene *L*_g_ is calculated to be 130 μm. In the 3D FDTD simulation, we set the length of the sidewall graphene as 128 μm, which is almost consistent with our theoretical calculation.

### 3.2. Generation and Switching of the Spin Angular Momentum (SAM)

In the first case, with the Fermi level setting at 0.9 eV for the bottom graphene, Figure 2 shows that the mode field energy of the quasi-TM mode is mainly localized in the low-refractive-index SiO_2_ between the bottom graphene and the GaAs core in the second waveguide section (GHPW), while the mode field distribution of the quasi-TE mode is almost the same as that transmitted in the first waveguide section. In addition, the effective refractive index difference (Δ*n*) between the two orthogonal modes in the second waveguide section is 0.179. With a phase difference of π/2 formed in the birefringent second section, the left-handed circularly polarized light is output into the third waveguide section (pure dielectric waveguide), which is validated by the vector field as shown in Figure 2c.

Similarly in the second case, as shown in Figure 3, when the Fermi level of the sidewall graphene is set with *E_f_* = 0.9 eV, it can be found that the mode field energy of the quasi-TE mode is mainly localized in the low-refractive-index SiO_2_ between the sidewall graphene and the GaAs core, while the mode field distribution of the quasi-TM mode is almost the same as that in the first waveguide section, and the effective refractive index difference (Δ*n*) of the two modes in the second waveguide section is −0.191. Due to the phase difference of −π/2 formed in the birefringent second section, the right-handed circularly polarized light is output into the third waveguide section, which is validated by the vector field, as shown in Figure 3c.

In short, due to the ultra-high confinement, the quasi-TE hybrid surface SPP mode and the quasi-TM hybrid SPP mode have higher effective refractive indices and propagation phases than those of the quasi-TE and quasi-TM (core) modes of a pure GaAs waveguide. Therefore, the left or right side graphene (or the bottom or top side graphene) can be used to modulate the phase of the quasi-TE light (or the quasi-TM light), for the generation of left or right circular polarization light with ±π/2. If deposited on all three (top, right, and left) sides other than two sides (top (or bottom) and right (or left)), the effect is similar but a bit complex. Therefore, in order to make it more convenient for sample fabrication, we chose the bottom and right sidewall graphene to modulate the two modes of light.

### 3.3. Generation and Switching of the Orbital Angular Momentum (OAM)

According to the previous discussion, after passing through the first and second waveguide sections, the output THz light in the third waveguide section is not a pure circularly polarized light. Crucially, there is a longitudinal electric field component *E*_Z_ due to the strong spin-orbit coupling in the proposed waveguide according to Equation (2). Our results show that when the transverse electric field component *E*_T_ carries SAM, meanwhile the longitudinal electric field component *E*_Z_ carries OAM with a spiral-phased wavefront. Equation (2) gives a π/2 phase difference between *E*_Z_ and *E*_T_ (assuming that *E*_T_ is a pure real value, then *E*_Z_ is a pure imaginary value). As mentioned above, it produces a phase difference Δ*δ* due to the birefringent effect in the second waveguide section. Therefore, there was the same phase difference of Δ*δ* in *E*_Z_ of the orthogonal modes. Figure 4 compares |*E*_Z_| formed in the first and third waveguide sections. In the first waveguide section, the phase difference between the two modes is nearly zero, so their component is antisymmetric with respect to an axis with *α* = 45°. After passing through the second waveguide section, Δ*δ* is ±π/2, and their coherently super-positioned field shows the first-order spiral phase, as shown in Figure 4b. The electric field distribution of the longitudinal component (*E*_Z_) in the third waveguide section can be expressed as:(6)$$E_{Z}(r,\theta )=A(r)e^{i\ell \theta },$$ where *A*(*r*), (*r*,*θ*), and *ℓ* are respectively the complex amplitude of the electric field, polar coordinate parameters, and the order of topological charge. Therefore, Fermi levels of graphene capacitors determine the polarization state of *E*_T_ and the wavefront of *E*_Z_ of the output light.

As shown in Figure 5, when the incident light is with a polarization angle *α* = 45°, the Fermi level *E_f_* of the bottom graphene is set as 0.9 eV, and the phase of the generated vortex beam is rotated anticlockwise by one cycle (*ℓ* = −1), alternatively, which is rotated clockwise by one cycle (*ℓ* = 1) when the Fermi level of the sidewall graphene is set as 0.9 eV.

### 3.4. Generation of Simultaneous Reversal of the Waveguide Mode Chirality and Spirality

According to the previous discussion, the left/right-handed circularly polarized light can be generated from a linearly polarized light with *α* = 45° using the birefringent effect of the GHPW. However, in many fields of on-chip nano-photonics, bio-photonics, and quantum information science, it is expected that the chiral and spiral conversion can be realized on chip without using bulk optical devices. By simply doubling the length of the birefringent waveguide section, a half wave plate (HWP) can be constructed, which can convert the left-hand quasi-circular polarization mode to the right-hand circular polarization mode on chip, and it is worth noting that the corresponding topological charge of the vortex field can also be reversed, and vice versa.

When the voltage is only applied to the bottom graphene with the Fermi level of 0.9 eV, and the length of the second waveguide section is extended to *L*_g_ = 270 μm, the resulting phase difference *δ* = π between the two modes is calculated by using Formula (5). Similarly, doubling the length of the sidewall graphene can also achieve the same effect (*δ* = -π). Figure 6a,b shows the simultaneous reversal of chirality and spirality (topological charge) of our device, which are validated by inversion of the rotating direction in the vector fields (|*E*_T_|) and inversion of the spiraling direction in phase distributions (*φ*_Z_), as shown in the insets.

### 3.5. Generation of Selective Linear Polarization Filtering

In addition, by adjusting the Fermi level of graphene, the GHPW can also be used for selective linear polarization filtering. Figure 7 shows the simulation results of the attenuation of the quasi-TE and the quasi-TM modes versus the Fermi level of bottom graphene or sidewall graphene. As shown in Figure 7a, when the voltage is applied only to the bottom graphene, the loss of the quasi-TM mode decreases with the increase of Fermi level of the bottom graphene, while the loss of the quasi-TE mode remains almost unchanged. When the Fermi levels of bottom graphene is 0.2 eV and sidewall graphene is 0 eV, attenuations of the quasi-TE mode and quasi-TM mode are respectively 6.78 and 42.07 dB/mm, and the extinction ratio is 35.29 dB/mm. The quasi-TE mode passes through and the quasi-TM mode is fully absorbed in the GHPW. Similarly, as shown in Figure 7b, when the voltage is only applied to the sidewall graphene, the loss of the quasi-TE mode decreases with the increase of the Fermi level of the sidewall graphene, while the loss of the quasi-TM mode remains on the same low-loss level. When the Fermi levels of sidewall graphene is 0.2 eV and the bottom graphene is 0 eV, attenuations of the quasi-TE mode and quasi-TM mode are respectively 52.26 and 6.64 dB/mm, and the extinction ratio is 45.26 dB/mm. Therefore, the waveguide can be reconfigured as a TM-mode pass and TE-mode blocked polarizer.

## 4. Conclusions

In summary, we have proposed and numerically demonstrated a reconfigurable graphene-based hybrid plasmonic waveguide (GHPW) with four different functions for on-chip AM manipulation and polarization filtering. Biasing one of the graphene capacitors can produce a switchable relative phase difference of ±π/2 between the decomposed quasi-TE and quasi-TM modes, which results from the strong waveguide birefringence between the guided mode and the selectively excited hybrid plasmonic mode. Therefore, the transverse component of the electric field of output beam possesses the SAM and longitudinal component possesses the OAM due to the spin-orbit coupling in the GHPW with switchable mode chirality and spirality. Based on the same principle, a simultaneous reversal of the waveguide mode chirality and spirality can be realized when the switchable relative phase difference is ±π. In addition, the selective linear polarization filtering with a high polarization extinction ratio can be achieved based on the polarization-dependent and biasing-dependent attenuation in the GHPW. Our proposed reconfigurable GHPW with multiple functions will greatly expand its applications in integrated quantum computing, on-chip bio-sensing, and terahertz communication.

## Figures and Tables

**Figure 1 nanomaterials-10-02436-f001:**
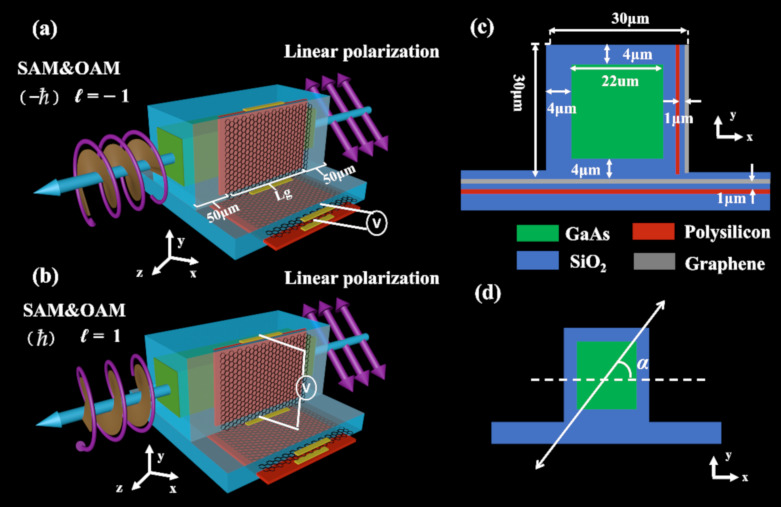
(**a**), (**b**) Schematic of the proposed reconfigurable terahertz (THz) graphene-based hybrid plasmonic waveguide (GHPW) for manipulating angular momentums of guided mode. The linearly polarized light with a polarization angle *α* = 45° can be converted into the output beam with spin angular momentum (SAM) = −*ħ* and orbital angular momentum (OAM) *ℓ* = −1 by biasing the bottom graphene capacitor (**a**) and the output beam with SAM = + *ħ* and OAM *ℓ* = +1 by biasing the sidewall graphene capacitor (**b**); (**c**) cross-section of the GHPW; (**d**) definition of the polarization angle α of a linearly polarized light incidence into the first dielectric waveguide section.

**Figure 2 nanomaterials-10-02436-f002:**
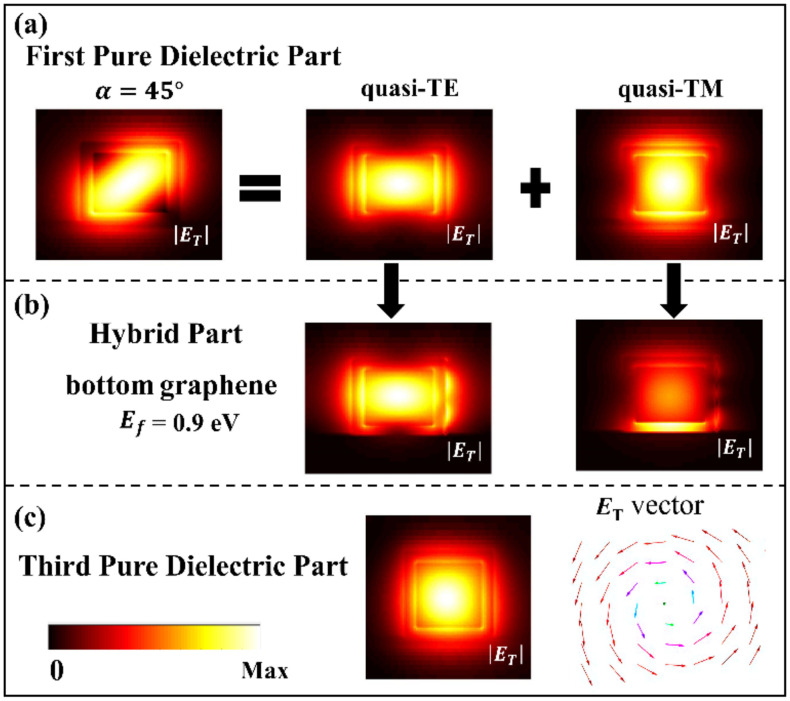
(**a**) Transverse component of the electric field (|*E*_T_|) of a linearly polarized light with *α* = 45° transmitted in the first waveguide section (pure dielectric waveguide) decomposed into |*E*_T_| of the quasi-TE mode and the quasi-TM mode. (**b**) |*E*_T_| of the quasi-TE mode and the quasi-TM mode transmitted in the second waveguide section (GHPW) when the Fermi level of the bottom graphene is set with *E_f_* = 0.9 eV. (**c**) |*E*_T_| and its vector field showing a generated left-handed circularly polarized (LCP) light (SAM = −*ħ*) transmitted in the third waveguide section (pure dielectric waveguide).

**Figure 3 nanomaterials-10-02436-f003:**
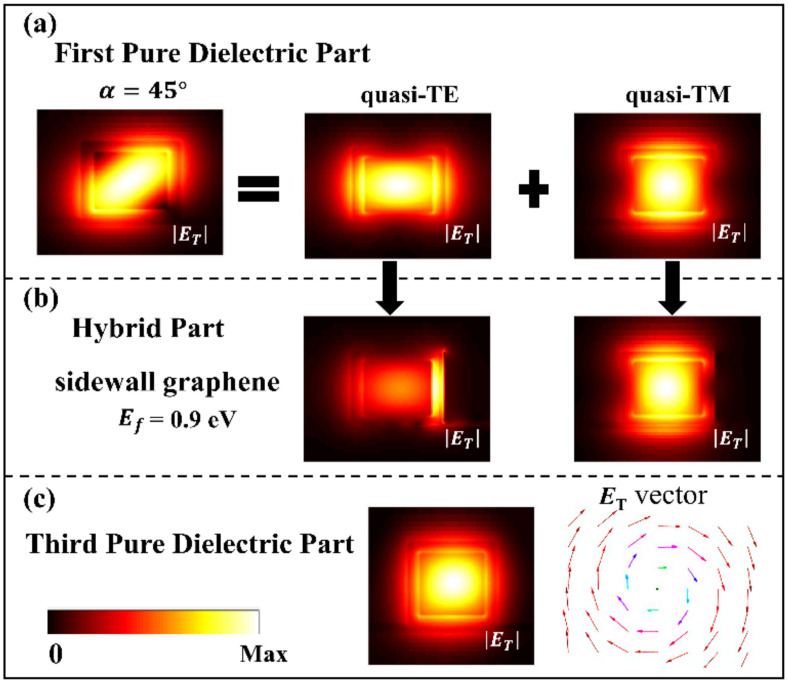
(**a**) Transverse component of the electric field (|*E*_T_|) of a linearly polarized light with *α* = 45° transmitted in the first waveguide section (pure dielectric waveguide) decomposed into |*E*_T_| of the quasi-TE mode and the quasi-TM mode. (**b**) |*E*_T_| of the quasi-TE mode and the quasi-TM mode transmitted in the second waveguide section (GHPW) when the Fermi level of the sidewall graphene is set with *E_f_* = 0.9 eV. (**c**) |*E*_T_| and its vector field showing a generated right-handed circularly polarized (RCP) light (SAM = +*ħ*) transmitted in the third waveguide section (pure dielectric waveguide).

**Figure 4 nanomaterials-10-02436-f004:**
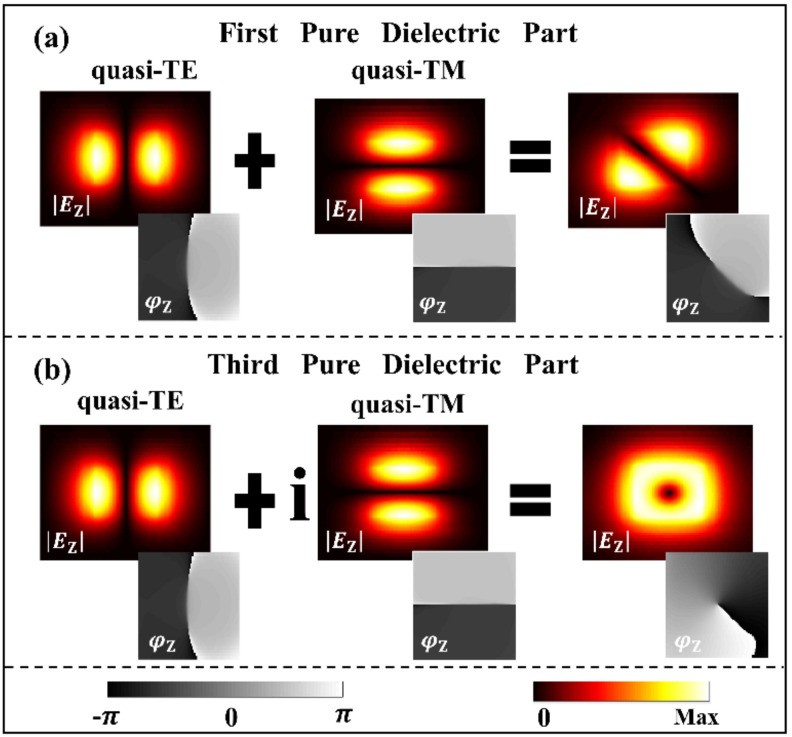
Longitudinal component of the electric fields (|*E*_Z_|) and phase distributions (φ_Z_) of the quasi-TE mode and the quasi-TM mode superimposed for the input linearly polarized beam with *α* = 45° transmitted in the first waveguide section (**a**) and for the generated output vertex beam in the third waveguide section when the Fermi level of the bottom graphene is set with *E_f_* = 0.9 eV and the Fermi level of the sidewall graphene is 0 eV (**b**), respectively.

**Figure 5 nanomaterials-10-02436-f005:**
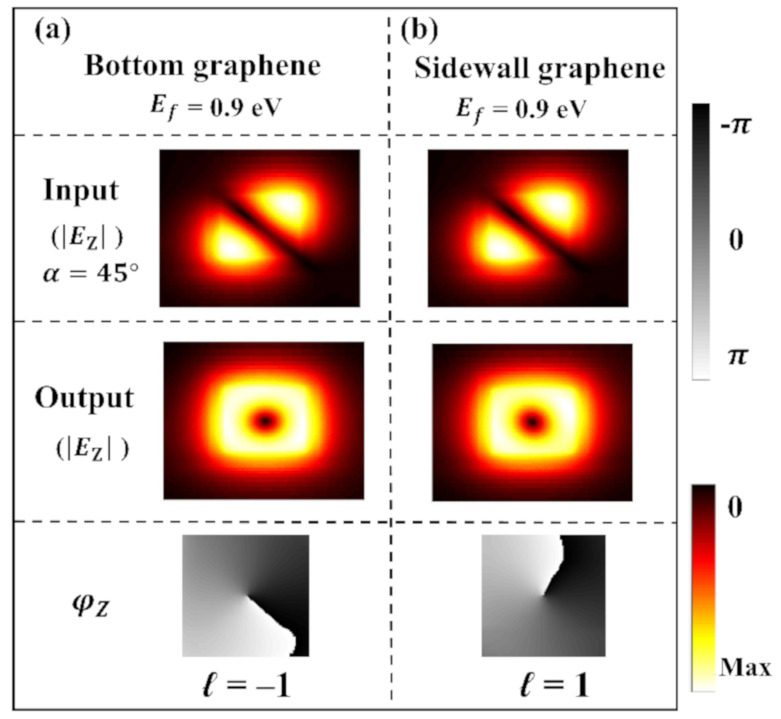
Longitudinal component of the electric fields (|*E*_Z_|) and their phase distributions (φ_Z_) for the input linearly polarized beam with α = 45° and the generated output vertex beams with switchable mode spirality when the Fermi level of bottom graphene is set with *E_f_* = 0.9 eV (**a**) and the Fermi level of sidewall graphene is set with *E_f_* = 0.9 eV (**b**).

**Figure 6 nanomaterials-10-02436-f006:**
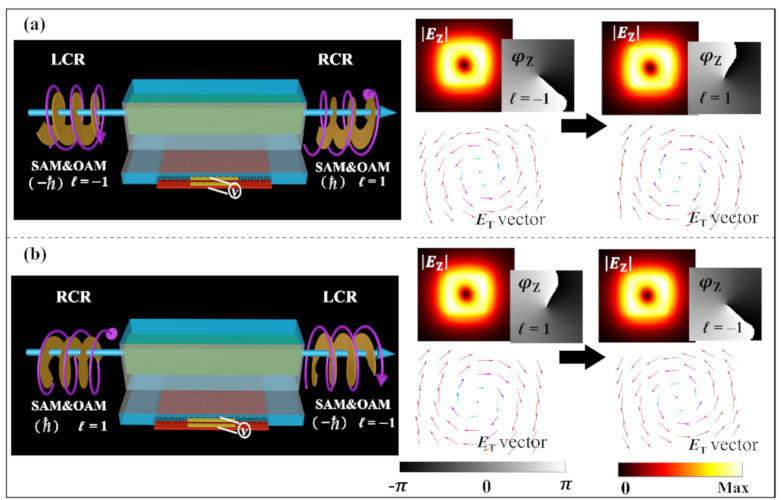
(**a**) Conversion of the RCP light into the LCP light. (**b**) Conversion of the LCP light into the RCP light. Simultaneous reversal of chirality and spirality (topological charge) of the waveguide modes are respectively validated by inversion of the rotating direction in the vector fields (|*E*_T_|) and phase distributions (φ_Z_), as shown in the insets.

**Figure 7 nanomaterials-10-02436-f007:**
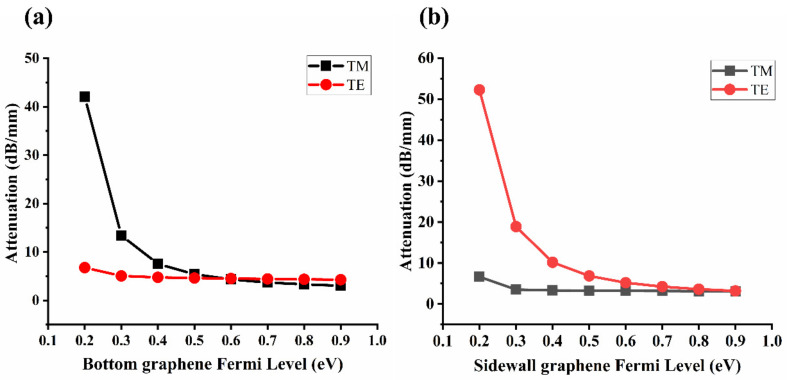
(**a**) Attenuation of the quasi-TE and the quasi-TM modes versus the Fermi level of bottom graphene and with zero biasing on the sidewall graphene. (**b**) Attenuation of the quasi-TE and the quasi-TM modes versus the Fermi level of the sidewall graphene and with zero biasing on the bottom graphene.

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
