# Peer review of "Active Manipulation of the Spin and Orbital Angular Momentums in a Terahertz Graphene-Based Hybrid Plasmonic Waveguide"

_nanomaterials, 2020, doi:10.3390/nano10122436_

Round 1
Reviewer 1 Report
1) Why graphene layer is deposited on one side? What will be the effect on the performance of the device, if deposited on all three (top, right, and left) sides?
2) Section 3.5, First line, “In addition” is repeated twice.
3) What effect can appear if the input light has an angle of greater or lower than 45 degrees?
4) What is the wavelength of the THz light used in the work? Please mention.
5) The dimensions of the waveguide are quite large. Can you please elaborate on the selection of such a big core dimension?
6) Section 2. The author has written that …….” waveguide core with a refractive index of 3.6 and with a length and width of 22 μm”…… I think the author has made a typo error. The height and width are 22 microns. The length is 50 microns.
7) Check for any English language-related issues.
Reviewer 2 Report
This manuscript presents graphen -based hybrid plasmonic waveguide for manipulating angular momentums of guided mode. The manuscript is written well and I can recommend it for publication in the revised form.
Comments: 1) It is not clear from the text of the manuscript whether the device described in the manuscript is really developed or it is only proposed. 2) In my opinion the authors of the scientific article can be only the scientists who made scientific research. The people who only wrote the paper but did no scientific investigations are not authors. Science is not literature. See "author contribution". 3) There is a typo in the beginning of Sec. 3.5. 4) What are shown in Fig 7, results of real measurements or results of numerical calculations? 5) Text in lines 270-277 repeats the text in lines 248-255Author Response
Please see the attachment
